# In Vitro Activity of Cefiderocol against a Global Collection of Carbapenem-Resistant *Acinetobacter baumannii* Isolates

**DOI:** 10.3390/antibiotics12071172

**Published:** 2023-07-11

**Authors:** Harald Seifert, Carina Müller, Danuta Stefanik, Paul G. Higgins, Esther Wohlfarth, Michael Kresken

**Affiliations:** 1Institute for Medical Microbiology, Immunology and Hygiene, Faculty of Medicine and University Hospital Cologne, University of Cologne, 50935 Cologne, Germanypaul.higgins@uni-koeln.de (P.G.H.); 2German Center for Infection Research (DZIF), Partner Site Bonn-Cologne, 38124 Braunschweig, Germany; 3Antiinfectives Intelligence GmbH, 51105 Cologne, Germany; esther.wohlfarth@antiinfectives-intelligence.de (E.W.); michael.kresken@antiinfectives-intelligence.de (M.K.); 4Rheinische Fachhochschule gGmbH, 50676 Cologne, Germany

**Keywords:** international clone, oxacillinase, β-lactamase inhibitor, colistin, multidrug resistance

## Abstract

**Background:** Cefiderocol is a novel siderophore cephalosporin with potent activity against multi-drug-resistant Gram-negative pathogens including carbapenem-resistant *Acinetobacter baumannii* (CRAB). **Methods:** The susceptibility of 313 non-duplicate CRAB isolates with defined carbapenem resistance mechanisms from a global collection to cefiderocol, ceftazidime, ceftazidime/avibactam, ceftolozane/tazobactam, ciprofloxacin, colistin, imipenem/relebactam, meropenem, meropenem/vaborbactam, minocycline, and piperacillin/tazobactam was determined using the broth microdilution method. Isolates were obtained from various body sites from patients in 47 countries in five world regions between 2012 and 2016. The identification of carbapenem resistance mechanisms and assignment to *A. baumannii* international clonal lineages were based on whole genome sequencing. **Results:** Cefiderocol showed greater activity than comparator antimicrobials of the β-lactam class, including novel β-lactams combined with β-lactamase inhibitors, ciprofloxacin, and minocycline. Cefiderocol MIC_50_ and MIC_90_ values were 0.5 mg/L and 4 mg/L, respectively, while colistin had comparable activity with a higher MIC_50_ at 1 mg/L and a lower MIC_90_ value of 2 mg/L. Many isolates with elevated cefiderocol MICs ≥ 4 mg/L represented *A. baumannii* international clone (IC) 1 and harbored a metallo-β-lactamase. **Conclusions:** While cefiderocol is a useful addition to the limited armamentarium of drugs targeting this problematic pathogen, a considerable part of CRAB isolates had elevated MIC values in a range of 4 -> 32 mg/L, including all isolates with a metallo-β-lactamase.

## 1. Introduction

*Acinetobacter baumannii* is a nosocomial pathogen that is notorious for its multidrug resistance and propensity to cause hospital outbreaks and epidemic spread [1,2]. *A. baumannii* isolates representing international clonal lineage (IC) 2 represent the majority of isolates recovered worldwide [3,4]. Multi-drug resistance and, in particular, resistance to carbapenems are matters of great concern. Carbapenem-resistant *A. baumannii* (CRAB) is considered as priority 1 (“critical”) in the WHO priority pathogens list for research, discovery, and development of new antibiotics published in 2017 [5], and it has recently been upgraded as an urgent public health threat by the CDC, emphasizing that, in the United States, annually, 8500 infections are caused by CRAB [6]. Resistance to carbapenems in *A. baumannii* is primarily caused by carbapenem-hydrolysing class D ß-lactamases (CHDLs) and, less frequently, metallo-β-lactamases [3,7]. Frequently, colistin is the only compound showing measurable activity against carbapenem-resistant *A. baumannii*. However, its therapeutic use is limited by toxicity and low-serum and tissue concentrations [8,9]. In addition, colistin resistance has been increasing, in particular in countries with heavy use of this drug for the treatment of infections caused by carbapenem-resistant *A. baumannii*, *Enterobacterales*, and *Pseudomonas aeruginosa* [10].

Several new drugs with activity against multi-drug-resistant (MDR) Gram-negative pathogens have been developed, including ceftolozane/tazobactam, ceftazidime/avibactam, plazomicin, imipenem/relebactam, and meropenem/vaborbactam, but these have only very limited activity against carbapenem-resistant *A. baumannii.* In a recent review, Shields et al. summarized the available treatment options for patients with CRAB infections [11]. Therefore, there is an urgent need to develop new drugs targeting serious infections caused by MDR *A. baumannii*. Cefiderocol, previously known as S-649266, is a novel catechol-substituted parenteral siderophore cephalosporin that has potent activity against a broad range of Gram-negative bacteria, including multi-drug-resistant pathogens such as carbapenem-resistant Enterobacterales and non-fermenting bacteria, such as *A. baumannii*, *P. aeruginosa*, and *Stenotrophomonas maltophilia* [12,13,14,15,16]. Cefiderocol was approved by the U.S. Food and Drug Administration (FDA) for the treatment of complicated urinary tract infections (cUTIs), including pyelonephritis, caused by susceptible Gram-negative microorganisms in adult patients who have limited or no alternative treatment options, and for the treatment of hospital-acquired and ventilator-associated bacterial pneumonia, caused by susceptible Gram-negative microorganisms. Cefiderocol has also been approved by the European Medicines Agency (EMA) for complicated Gram-negative infections in adults with limited treatment options. It has been shown to possess high stability against various β-lactamases, including carbapenem-hydrolysing class A β-lactamases, such as KPC; class B metallo-β-lactamases (MBLs), IMP, VIM, and NDM; and carbapenem-hydrolysing class D β-lactamases, such as OXA-48 and OXA-23, and it is the only new compound with good activity against CRAB that has entered the market [17,18].

In a randomized registrational trial in patients with Gram-negative nosocomial pneumonia, in 16% of cases, pneumonia was caused by *A. baumannii*, and cefiderocol was non-inferior to meropenem in terms of all-cause mortality and was considered a potential option for the treatment of patients with nosocomial pneumonia including those caused by MDR Gram-negative bacteria [19]. In another trial comparing cefiderocol to the best available therapy (BAT) in critically ill patients with nosocomial pneumonia (NP), bloodstream infections (BSIs), sepsis, or cUTIs and evidence of a carbapenem-resistant Gram-negative pathogen, 46% of which were CRAB, cefiderocol had similar clinical and microbiological efficacy to BAT. However, an increase in all-cause mortality (34% vs. 18%) at the end of the study was observed in patients treated with cefiderocol, as compared to BAT [20]. Generally, deaths were more frequent in patients with infections caused by *A. baumannii* than by other pathogens (49% vs. 18%). The cause of the increase in mortality has not been established. In contrast, in a recent observational retrospective study comparing cefiderocol to colistin-based regimens in the treatment of severe infections caused by CRAB, 30-day mortality was higher in patients receiving colistin compared to those receiving cefiderocol-containing regimens (56% vs. 34%, *p* = 0.018) [21]. This difference was confirmed in patients with BSI but not in those with NP, which showed equivalent mortality to colistin-based therapy. Of note, among eight cases in the cefiderocol group who experienced microbiological failure, four (50%) developed resistance to cefiderocol.

The purpose of this study was to evaluate the activity of cefiderocol in comparison to reference antimicrobials, including novel β-lactam-β-lactamase inhibitor combinations against a collection of well-characterized, non-duplicate, global isolates of carbapenem-resistant *A. baumannii* (CRAB) harboring acquired oxacillinases, metallo-β-lactamases, or encoding an up-regulated intrinsic OXA-51-like carbapenemase.

## 2. Results

Among the 313 CRAB isolates, 234 isolates harbored *bla*_OXA-23-like_, and 56 isolates had *bla*_OXA-40-like_. Other carbapenemases are listed in Table 1. Based on core genome MLST results, isolates represented the eight previously described major international clonal lineages IC1 (N = 26), IC2 (196), IC3 (2), IC4 (5), IC5 (44), IC6 (3), IC7 (12), and IC8 (5), while 20 isolates did not cluster with any of the international clonal lineages [3].

Table 2 shows the MIC distribution, MIC_50_ and MIC_90_ values, MIC ranges, and percent susceptibility. All isolates were resistant to imipenem/relebactam and non-susceptible to meropenem; 99% of isolates were resistant to ciprofloxacin and 4.8% to colistin.

The cefiderocol MIC_50/90_ values were 0.5 and 4 mg/L, respectively, with 18.2% of isolates being resistant when applying EUCAST non-species-related PK-PD breakpoints (susceptible, ≤2 mg/L; resistant, >2 mg/L) [22]. Of note, if the recently published CLSI breakpoints had been applied (susceptible, ≤4 mg/L; resistant, ≥16 mg/L), resistance to cefiderocol would drop to 5.1% [23].

Comparatively, MIC_50/90_ values were ≥64/≥64 mg/L for ceftazidime; ≥16/4/≥16/4 mg/L for ceftazidime/avibactam; ≥16/4/≥16/4 mg/L for ceftolozane/tazobactam; ≥8/≥8 mg/L for ciprofloxacin; 1/2 for colistin; ≥16/4/≥16/4 mg/L for imipenem/relebactam; ≥32/≥32 mg/L for meropenem; ≥16/8/≥16/8 mg/L for meropenem/vaborbactam; 4/≥16 mg/L for minocycline; and ≥128/4/≥128/4 mg/L for piperacillin/tazobactam. Among the fifteen isolates that were resistant to colistin, two were also resistant to cefiderocol, with MIC values of ≥64 mg/L.

We did not find a correlation between the major *bla*_OXA_ types and cefiderocol MICs, as depicted in Table 3. The cefiderocol MIC_50/90_ was 0.5/4 mg/L for isolates with OXA-23-like (N = 227) and only slightly lower with 0.5/1 mg/L for isolates with OXA-40-like (N = 54). However, resistance to cefiderocol was considerably higher at 40.6% in *A. baumannii* isolates with other carbapenemases, with MIC_50/90_ values of 1 and 32 mg/L, respectively (Table 3 and Table 4). All seven *A. baumannii* isolates harboring the metallo-β-lactamase *bla*_NDM-1_ (including one isolate that co-harbored *bla*_OXA-23_, *bla*_NDM-1_ and *bla*_PER-7_) or *bla*_IMP-26,_ as well as three of four isolates harboring a GES-type β-lactamase, exhibited cefiderocol MICs ≥4 mg/L and were considered resistant according to EUCAST breakpoints (Table 4).

Of interest, *A. baumannii* isolates representing IC1 (N = 26) had higher cefiderocol MIC_90_ values (32 mg/L) than isolates representing other clonal strain types, such as IC2 (N = 196; MIC_90_, 4 mg/L) and IC5 (N = 44; MIC_90_, 1 mg/L), resulting in a higher resistance rate of 30.8% among *A. baumannii* IC1 isolates compared to 16.3% and 4.5% for the two other clonal lineages IC2 and IC5 (Table 5). There were no differences between international clonal lineages with regard to MIC_90_ values of the other antimicrobials tested.

## 3. Discussion

In our study, we assessed the in vitro activity of cefiderocol against a panel of carbapenem-resistant *A. baumannii* representing a unique worldwide collection of isolates. Given the well-known clonal population structure of *A. baumannii* and their tendency for hospital outbreaks and endemic persistence, every effort was made to ensure the greatest possible strain diversity. WGS was used to identify currently circulating international clonal lineages and carbapenem resistance mechanisms.

We found that cefiderocol had potent activity against carbapenem-resistant *A. baumannii* isolates with MIC_50/90_ values of 0.5 and 4 mg/L, respectively. When applying the non-specific PK-PD breakpoints (susceptible, ≤2; resistant >2 mg/L) provided by EUCAST, 18.2% of isolates were classified as resistant. Conversely, when the recently published CLSI breakpoints (susceptible, ≤4 mg/L; resistant, ≥16 mg/L) were applied, only 5.1% of isolates were considered resistant. Among the antimicrobial agents tested, only colistin had comparable activity, while, as expected, novel β-lactam-β-lactamase inhibitors were not active against carbapenem-resistant *A. baumannii.* Hackel et al., in a collection of meropenem-non-susceptible *A. baumannii* isolates from the US and Europe from 2014 to 2015, found comparable activity with slightly lower cefiderocol MIC_50/90_ values of 0.25 and 1 mg/L, respectively [14].

Our data also concur with findings reported by Shortridge et al., in a more recent surveillance study of meropenem-resistant *A. baumannii* isolates collected in 2020, who reported cefiderocol MIC_50/90_ values of 0.25 and 2 mg/L, respectively, and a resistance rate of 2.9%, with no difference between isolates collected in the US and in Europe [15]. However, in the latter studies, no information regarding the epidemiological background and resistance mechanisms of the isolates under study was provided. In contrast, Ballesté-Delpierre et al., in a more limited collection of 113 *A. baumannii* isolates from nine countries, recently reported a much higher cefiderocol MIC_90_ value of >64 mg/L and a considerably higher resistance rate of 20.4% using CLSI breakpoints [24]. These differences may be explained by a less well-balanced strain collection, including a large number of isolates from Azerbaijan, that was highly resistant to cefiderocol.

When different *A. baumannii* clonal strain types were compared, isolates representing IC1 had higher cefiderocol MIC_90_ values and higher resistance rates than isolates assigned to other major lineages, such as IC2 and the Pan-American lineage IC5. We currently have no explanation for this finding. Conversely, Ballesté-Delpierre et al. reported a weak association of resistance to cefiderocol with IC2 isolates [24].

There was no correlation between specific *bla*_OXA_ genes and elevated cefiderocol MICs. Conversely, all isolates harboring metallo-β-lactamases and the majority of isolates harboring GES-like β-lactamases in addition to oxacillinases had cefiderocol MIC values of ≥4 mg/L. This finding concurs with recently reported data that NDM-like β-lactamases and, to an even greater extent, PER-like β-lactamases were found to be associated with reduced susceptibility to cefiderocol in CRAB. Poirel et al. found cefiderocol MIC values ranging from 2 to 16 mg/L among eight *bla*_NDM_-positive *A. baumannii* isolates [25]. Resistance to cefiderocol has also been found to be associated with the presence of *bla*_PER-like_ [25,26]. Poirel and colleagues found 8 isolates among a collection of 87 *A. baumannii* isolates with cefiderocol MIC values of ≥8 mg/L, all of them harboring *bla*_OXA-23_ and either *bla*_PER-1_ or *bla*_PER-7_ [25]. The authors also showed a ≥16-fold increase in cefiderocol MIC values after the *bla*_PER-1_ gene was cloned into a shuttle plasmid and subsequently electroporated into the *A. baumannii* CIP70.10 recipient strain, thus confirming the contribution of *bla*_PER-1_ to cefiderocol resistance. Liu and colleagues claimed the production of PER-1 to be the key mechanism of cefiderocol resistance in 131 *A. baumannii* isolates from China and found that PER-1 could be inhibited by a combination of cefiderocol and avibactam or durlo-bactam [26]. However, in our global collection, *bla*_PER-like_ was found in only one isolate and in combination with *bla*_OXA-23_ and *bla*_NDM-1_, so its real impact on cefiderocol resistance from a global perspective remains unclear. In fact, the majority of our isolates exhibiting elevated cefiderocol MIC values harbored neither PER-type nor NDM-type β-lactamases; thus, the major cefiderocol resistance mechanism in *A. baumannii* still needs to be elucidated.

Our study has several limitations. First, the applicability of the non-specific PK-PD breakpoints provided by EUCAST for *A. baumannii* is problematic, in particular, as the EUCAST breakpoint document states that there is insufficient evidence that *A. baumannii* is a good target for therapy with cefiderocol [22]. On the other hand, there is no such caveat mentioned in the CLSI document [23]. Second, our strain collection dates back to 2016, and a shift in both strain-type prevalence and resistance mechanisms since then is possible.

In conclusion, cefiderocol had good in vitro potency against carbapenem-resistant *A. baumannii*, including isolates that were resistant to colistin, and, thus, may be a promising therapeutic option for the treatment of infections due to MDR *A. baumannii*. However, we found considerable resistance among our isolates, in particular, but not only, in isolates harboring metallo-β-lactamases. This could be a cause for concern as there is a high medical need, particularly for *A. baumannii* infections caused by metallo-β-lactamase, to produce isolates that are not susceptible to other novel antimicrobials with activity against MDR *A. baumannii* such as sulbactam-durlobactam [27].

## 4. Materials and Methods

### 4.1. Bacterial Isolates

The *A. baumannii* isolates included in our study were obtained between 2012 and 2016 from various body sites in patients from 114 hospitals in 47 countries and from five world regions, Africa (N = 29), Asia and Middle East (88), Europe (67), Latin America (76), and North America (53). The average number of isolates per hospital was 2.7, with only 15% of hospitals contributing more than 3 isolates over the 5-year study period (copy strains were excluded). To optimally reflect the current global epidemiology of carbapenem-resistant *A. baumannii,* the number of isolates included was based on the population size of participating countries. The isolates were subjected to whole genome sequencing (WGS) using the Illumina MiSeq platform, and MLST types were derived from WGS data. Carbapenem resistance mechanisms were determined as described previously. The presence of oxacillinase-encoding genes (*bla*_OXA-51-like,_
*bla*_OXA-23-like_, *bla*_OXA-40-like_, *bla*_OXA-58-like,_
*bla*_OXA-143-like_ and *bla*_OXA-235-like_) was investigated using a previously described multiplex PCR [28,29]. Two further multiplex PCRs were applied to detect *bla*_VIM_, *bla*_KPC_, *bla*_NDM_, *bla*_IMI_, *bla*_GES_, *bla*_GIM_, *bla*_IMP,_ and IS*Aba1* upstream of *bla*_OXA-51-like_ [30]. In addition, ResFinder 3.1 (https://cge.cbs.dtu.dk/services/ResFinder/ accessed on 1 May 2023) was applied to determine the acquired resistome of each isolate from sequencing data with special interest in the distinct variants of the carbapenemase families identified using PCR. The raw sequencing reads generated in this project were submitted to the European Nucleotide Archive (https://www.ebi.ac.uk/ena/ accessed on 1 May 2023) under the study accession number PRJEB27899.

### 4.2. Antimicrobial Susceptibility Testing

Antimicrobial susceptibility testing was performed using broth microdilution in cation-adjusted Mueller–Hinton broth (CAMHB) according to the standard ISO 20776-1 [31]. For cefiderocol testing, iron-depleted CAMHB (iron concentration, ≤0.03 mg/L) was used [23]. In-house-prepared 96-well plates were used for cefiderocol, and 96-well plates, supplied by International Health Management Associates Inc. (Schaumburg, IL, USA), were used for reference antimicrobials. Three hundred and thirteen non-duplicate carbapenem-resistant *A. baumannii* isolates were tested against cefiderocol, ceftazidime, ceftazidime/avibactam, ceftolozane/tazobactam, ciprofloxacin, colistin, imipenem/ relebactam, meropenem, meropenem/vaborbactam, minocycline, and piperacillin/tazobactam. Resistance to the carbapenems imipenem and meropenem has been previously confirmed by Etest (bioMérieux, Nürtingen, Germany). The concentration ranges tested in two-fold dilutions were cefiderocol, 0.03–32 mg/L; ceftazidime, 0.03–32 mg/L; ceftazidime/avibactam, 0.12/4–8/4 mg/L; ceftolozane/tazobactam, 0.25/4–8/4 mg/L; cipro-floxacin, 0.002–4 mg/L; colistin, 0.12–4 mg/L; imipenem/relebactam, 0.03/4–8/4 mg/L; meropenem, 0.06–16 mg/L; meropenem/vaborbactam, 0.06/8–8/8 mg/L; minocycline, 0.12–8 mg/L; and piperacillin/tazobactam, 0.5/4–64/4 mg/L. MICs were interpreted following EUCAST guidelines, and susceptibility rates were determined using EUCAST breakpoints where applicable [22]. For cefiderocol, the EUCAST non-species-related PK-PD breakpoints were applied [22]. *Escherichia coli* ATCC 25922, *P. aeruginosa* ATCC 27853, and *A. baumannii* NCTC 13304 were used as quality control strains.

## Figures and Tables

**Table 1 antibiotics-12-01172-t001:** Distribution of carbapenemases among 313 carbapenem-resistant *A. baumannii* isolates.

Carbapenem Resistance Mechanism	No. of Isolates with Respective Mechanism ^a^
*bla* _OXA-23-like_	234
*bla* _OXA-40-like_	56
*bla* _OXA-58-like_	6
*bla* _OXA-235-like_	2
*bla* _NDM-1_	6
*bla* _IMP-26_	1
upregulated *bla*_OXA-51_	13

^a^ Of the isolates, 2 isolates co-harbored *bla*_OXA-23_ and *bla*_NDM-1_, 1 isolate co-harbored *bla*_OXA-23_, *bla*_NDM-1_, and *bla*_PER-7_, 3 isolates, *bla*_OXA-23_ and *bla*_GES-11_, 1 isolate, *bla*_OXA-23_ and *bla*_GES-12_, 1 isolate, *bla*_OXA-40_ and *bla*_GES-11_, 1 isolate, *bla*_OXA-23_ and *bla*_OXA-40_, and 1 isolate, *bla*_OXA-23_ and *bla*_OXA-58_.

**Table 2 antibiotics-12-01172-t002:** MIC distributions, MIC_50_ and MIC_90_ values (mg/L), and antimicrobial susceptibilities of 313 carbapenem-resistant *A. baumannii* isolates.

Antimicrobial Agent	MIC (mg/L)																	
	0.06	0.125	0.25	0.5	1	2	4	8	16	32	64	128	MIC_50_	MIC_90_	MIC range	%S	%I	%R
Cefiderocol ^a,b^	17	49	16	128	38	8 ^c^	32	9	1	5	10 ^d^		0.5	4	0.06 -≥ 64	81.8	-	18.2
Ceftazidime ^a^						1	7	6	6	16	277 ^d^		≥64	≥64	2 -≥ 64	-	-	-
Ceftazidime/avibactam ^a,e^						2	7	14	290 ^f^				≥16/4	≥16/4	2/4 -≥ 16/4	-	-	-
Ceftolozane/tazobactam ^a,e^					2	5	8	33	265 ^f^				≥16/4	≥16/4	1/4 -≥ 16/4	-	-	-
Ciprofloxacin		3				**0** ^g^	1	309 ^h^					≥8	≥8	0.125 -≥ 8	0.0	1.0	99.0
Colistin				8	208	**82** ^c^	1	14 ^h^					1	2	0.5 -≥ 8	95.2	-	4.8
Imipenem/relebactam ^e^						**0** ^c^		16	297 ^f^				≥16/4	≥16/4	8/4 -≥ 16/4	0.0	-	100.0
Meropenem ^l^						**0** ^c^		11	16	286 ^i^			≥32	≥32	8 -≥ 32	0.0	3.5	96.5
Meropenem/vaborbactam ^a,e^								15	298 ^f^				≥16/8	≥16/8	8/8 -≥ 16/8	-	-	-
Minocycline ^a^		2 ^k^	5	25	34	52	46	51	98 ^f^				4	≥16	≤0.125 -≥ 16	-	-	-
Piperacillin/tazobactam ^a,e^								1	1	1	1	309 ^j^	≥128/4	≥128/4	8 -≥ 128/4	-	-	-

^a^ no EUCAST breakpoint available; ^b^ for cefiderocol, the EUCAST non-species related PK-PD susceptible breakpoint is applied; ^c^ susceptible breakpoint values are indicated in boldface; ^d^ ≥64 mg/L; ^e^ for ceftazidime/avibactam, ceftolozane/tazobactam, imipenem/relebactam, and meropenem/vaborbactam, only the respective β-lactam compound concentration is given; ^f^ ≥16 mg/L; ^g^ for ciprofloxacin, the resistant breakpoint is depicted in bold face; ^h^ ≥8 mg/L; ^i^ ≥32 mg/L; ^j^ ≥128 mg/L; ^k^ ≤0.125 mg/L; ^l^ for meropenem, the non-meningitis breakpoint is applied.

**Table 3 antibiotics-12-01172-t003:** MIC_50_ and MIC_90_ values (mg/L) and antimicrobial susceptibilities of 313 carba-penem-resistant *A. baumannii* isolates harboring different carbapenemases.

Antimicrobial Agent	*bla_OXA-23-like_* (n = 227)					*bla_OXA-40-like_* (n = 54)					Other Carbapenemases (n = 32) ^d^				
	MIC_50_	MIC_90_	MIC range	%S	%R	MIC_50_	MIC_90_	MIC range	%S	%R	MIC_50_	MIC_90_	MIC range	%S	%R
Cefiderocol ^a,b^	0.5	4	0.06 -≥ 64	82.4	17.6	0.5	1	0.06 -≥ 64	92.6	7.4	1	32	0.06 -≥ 64	59.4	40.6
Ceftazidime ^a^	≥64	≥64	2 -≥ 64	-	-	≥64	≥64	4 -≥ 64	-	-	≥64	≥64	4 -≥ 64	-	-
Ceftazidime/avibactam ^a^	≥16/4	≥16/4	2/4 -≥ 16/4	-	-	≥16/4	≥16/4	2/4 -≥ 16/4	-	-	≥16/4	≥16/4	8/4 -≥ 16/4	-	-
Ceftolozane/tazobactam ^a^	≥16/4	≥16/4	1/4 -≥ 16/4	-	-	≥16/4	≥16/4	2/4 -≥ 16/4	-	-	≥16/4	≥16/4	2/4 -≥ 16/4	-	-
Ciprofloxacin	≥8	≥8	0.12 -≥ 8	0.4 ^c^	99.6	≥8	≥8	0.12 -≥ 8	1.9 ^c^	98.1	≥8	≥8	≥8 -≥ 8	0.0	100.0
Colistin	1	2	0.5 -≥ 8	96.0	4.0	1	2	0.5 -≥ 8	92.6	7.4	1	2	0.5 -≥ 8	93.8	6.2
Imipenem/relebactam	≥16/4	≥16/4	8/4 -≥ 16/4	0.0	100.0	≥16/4	≥16/4	8/4 -≥ 16/4	0.0	100.0	16/4	≥16/4	8/4 -≥ 16/4	0.0	100.0
Meropenem	≥32	≥32	8 -≥ 32	0.0	99.6	≥32	≥32	8 -≥ 32	0.0	98.1	16	≥32	8 -≥ 32	0.0	71.9
Meropenem/vaborbactam ^a^	≥16/8	≥16/8	8/8 -≥ 16/8	-	-	≥16/8	≥16/8	8/4 -≥ 16/8	-	-	≥16/8	≥16/8	8/4 -≥ 16/8	-	-
Minocycline ^a^	8	≥16	0.25 -≥ 16	-	-	4	≥16	≤0.12 -≥ 16	-	-	2	≥16	≤0.12 -≥ 16	-	-
Piperacillin/tazobactam ^a^	≥128/4	≥128/4	≥128/4 -≥ 128/4	-	-	≥128/4	≥128/4	8/4 -≥ 128/4	-	-	≥128/4	≥128/4	16/4 -≥ 128/4	-	-

^a^ no EUCAST breakpoint available; ^b^ for cefiderocol, the EUCAST non-species related PK-PD susceptible breakpoint is applied; ^c^ for ciprofloxacin, % intermediate is depicted. ^d^ other carbapenemases were *bla*_OXA-58-like_ (5 isolates); *bla*_OXA-235-like_ (2), *bla*_NDM-1_ (3); *bla*_IMP-26_ (1); and combinations of two or more carbapenemases including *bla*_OXA-23+NDM-1_ (2), *bla*_NOXA-23+NDM-1+PER-7_ (1), *bla*_NOXA-23+GES-11_ (3), *bla*_NOXA-23+GES-12_ (1), and *bla*_NOXA-40-like+GES-11_ (1); and upregulated *bla*_OXA-51_ (13).

**Table 4 antibiotics-12-01172-t004:** Distribution of carbapenem resistance mechanisms in 57 carbapenem-resistant *A. baumannii* isolates with cefiderocol MICs ≥ 4 mg/L.

Cefiderocol MIC	No. of Isolates with Respective MIC	Carbapenem Resistance Mechanism	No. of Isolates with Respective Mechanism
4 mg/L	32	upregulated *bla*_OXA-51_	2
		*bla* _OXA-23_	16
		*bla*_OXA-23_ + upregulated *bla*_OXA-51_	7
		*bla* _OXA-23+GES-11_	1
		*bla* _OXA-23+GES-12_	1
		*bla* _OXA-40-like_	3
		*bla* _NDM-1_	1
		*bla* _OXA-23+NDM-1_	1
8 mg/L	9	*bla* _OXA-23_	7
		*bla* _NDM-1_	1
		*bla* _IMP-26_	1
16 mg/L	1	*bla* _OXA-23_	1
32 mg/L	5	*bla* _OXA-23_	1
		*bla* _OXA-40-like+GES-11_	1
		upregulated *bla*_OXA-51_	1
		*bla* _OXA-23+NDM-1+PER-7_	1
		*bla* _NDM-1_	1
>32 mg/L	10	*bla* _OXA-23_	8
		*bla* _OXA-40-like_	1
		*bla* _OXA-23+NDM-1_	1

**Table 5 antibiotics-12-01172-t005:** MIC_50_ and MIC_90_ values (mg/L) and antimicrobial susceptibilities of 266 carba-penem-resistant *A. baumannii* isolates representing the most frequent international clonal lineages IC 1, IC 2, and IC 5.

Antimicrobial Agent	IC 1 (n = 26)					IC 2 (n = 196)					IC 5 (n = 44)				
	MIC_50_	MIC_90_	MIC range	%S	%R	MIC_50_	MIC_90_	MIC range	%S	%R	MIC_50_	MIC_90_	MIC range	%S	%R
Cefiderocol ^a,b^	0.5	32	0.06 -≥ 64	69.2	30.8	0.5	4	0.06 -≥ 64	83.7	16.3	0.5	1	0.06–4	95.5	4.5
Ceftazidime ^a^	≥64	≥64	4 -≥ 64	-	-	≥64	≥64	8 -≥ 64	-	-	≥64	≥64	4 -≥ 64	-	-
Ceftazidime/avibactam ^a^	≥16/4	≥16/4	4/4 -≥ 16/4	-	-	≥16/4	≥16/4	4/4 -≥ 16/4	-	-	≥16/4	≥16/4	2/4 -≥ 16/4	-	-
Ceftolozane/tazobactam ^a^	≥16/4	≥16/4	2/4 -≥ 16/4	-	-	≥16/4	≥16/4	4/4 -≥ 16/4	-	-	≥16/4	≥16/4	1/4 -≥ 16/4	-	-
Ciprofloxacin	≥8	≥8	4 -≥ 8	0.0	100.0	≥8	≥8	≥8 -≥ 8	0.0	100.0	≥8	≥8	≥8 -≥ 8	0.0	100.0
Colistin	1	2	0.5 -≥ 8	96.2	3.8	1	2	0.5 -≥ 8	94.9	5.1	2	2	1 -≥ 8	93.2	6.8
Imipenem/relebactam	≥16/4	≥16/4	8/4 -≥ 16/4	0.0	100.0	≥16/4	≥16/4	8/4 -≥ 16/4	0.0	100.0	16/4	≥16/4	8/4 -≥ 16/4	0.0	100.0
Meropenem	≥32	≥32	8 -≥ 32	0.0	96.2	≥32	≥32	8 -≥ 32	0.0	95.9	≥32	≥32	16 -≥ 32	0.0	100.0
Meropenem/vaborbactam ^a^	≥16/8	≥16/8	8/8 -≥ 16/8	-	-	≥16/8	≥16/8	8/8 -≥ 16/8	-	-	≥16/8	≥16/8	≥16/8 -≥ 16/8	-	-
Minocycline ^a^	2	4	0.25 -≥ 16	-	-	8	≥16	0.5 -≥ 16	-	-	1	4	0.25–8	-	-
Piperacillin/tazobactam ^a^	≥128/4	≥128/4	16/4 -≥ 128/4	-	-	≥128/4	≥128/4	32/4 -≥ 128/4	-	-	≥128/4	≥128/4	≥128/4 -≥ 128/4	-	-

^a^ no EUCAST breakpoint available; ^b^ for cefiderocol, the EUCAST non-species related PK-PD susceptible breakpoint is applied.

## Data Availability

Not applicable.

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
