# Peer review of "In Vitro Activity of Cefiderocol against a Global Collection of Carbapenem-Resistant Acinetobacter baumannii Isolates"

_antibiotics, 2023, doi:10.3390/antibiotics12071172_

Round 1

Reviewer 1 Report

Thank you for the opportunity to review this manuscript. In this study, the authors provided information on in vitro susceptibility of A. baumannii to cefiderocol and key beta-lactams, as well as a genomic background that may involve in the resistance. Overall, the study is interesting and the methods are sound. I only have one question about the methods. The remaining comments are about the data presentation.

Methods

As all A. baumannii already underwent WGS, why did the authors also perform PCR to identify resistance genes? I believe ResFinder should be sufficient.

Presentation

1.      In Table 1, the MIC distribution (and breakpoint) can be shown better using a histogram.

2.      In Table 2, the group with other carbapenemases have a significantly higher resistance prevalence. Can the author further explore which carbapenemase is associated with higher resistance?

3.      Table 3 does not provide much information. Instead, I think the first paragraph of the results can be presented better in Table format. Please consider this.

Author Response

Response to Reviewer 1

Thank you for the opportunity to review this manuscript. In this study, the authors provided information on in vitro susceptibility of A. baumannii to cefiderocol and key beta-lactams, as well as a genomic background that may involve in the resistance. Overall, the study is interesting and the methods are sound. I only have one question about the methods. The remaining comments are about the data presentation.

Methods

As all A. baumannii already underwent WGS, why did the authors also perform PCR to identify resistance genes? I believe ResFinder should be sufficient.

Response: This is a good question. We routinely use PCR because we have found that sometimes beta-lactamases do not assemble. By doing a pre-PCR we know in advance that an isolate has a particular carbapenemase.

Presentation

  1. In Table 1, the MIC distribution (and breakpoint) can be shown better using a histogram.

Response: The reviewer is right, that MIC distributions can be shown as a histogram. However, in the majority of scientific papers MIC distributions are shown in a table format. We therefore prefer not to change the presentation of the MIC data.

  1. In Table 2, the group with other carbapenemases have a significantly higher resistance prevalence. Can the author further explore which carbapenemase is associated with higher resistance?

Response: The requested information is given in the text (lines 136-139), in the footnotes to table 2 (now table 3), as well as in table 3 (now table 4).

  1. Table 3 does not provide much information. Instead, I think the first paragraph of the results can be presented better in Table format. Please consider this.

Response: The first paragraph of the results is presented in a table format as suggested. Table 3 (now table 4), however, is retained because this table contains the information requested by the reviewer in bullet point 2 above.

Reviewer 2 Report

The current manuscript is an interesting study on the in vitro antibacterial potential of a new drug molecule, cefiderocol, specifically against carbapenem-resistant Acinetobacter baumannii. The methodology appears to be sound and the results are reasonably adequately discussed. Nevertheless, some changes should be made before acceptance for publication:

- In the Introduction section, and in section 4.1, there are websites as a references, with active links, these should be removed and inserted properly like any other reference;

- An abbreviation list is missing;

- A figure should be made regarding current therapies, marketed and in development, against Carbapenem-resistant Acinetobacter baumannii Isolates, and their limitations;

- The fact that the studies were only done in vitro, and not in vivo, or in a clinical setting, should be mentioned and discussed as limitations of the study;

- In spite of the promising results, the authors still found bacterial resistance in specific cases; strategies to overcome this should be discussed, such as, for example, drug molecule functionalization/modification;

- The methods section should have a subsection describing the used statistical software and respective methodology;

- Given the general characteristics of the most promising molecules (lipophilicity/hydrophilicity, vulnerability do metabolism, etc.), what types of formulations would be best - Conventional? Nanosystems? And why; also what administration routes would be best?

Author Response

Response to Reviewer 2

The current manuscript is an interesting study on the in vitro antibacterial potential of a new drug molecule, cefiderocol, specifically against carbapenem-resistant Acinetobacter baumannii. The methodology appears to be sound and the results are reasonably adequately discussed. Nevertheless, some changes should be made before acceptance for publication:

- In the Introduction section, and in section 4.1, there are websites as references, with active links, these should be removed and inserted properly like any other reference;

Response: Done

- An abbreviation list is missing;

Response: Following the Author Guidelines of this journal we defined Acronyms/Abbreviations the first time they appear in each of three sections: the abstract; the main text; the first figure or table. We therefore stay with the current definitions of abbreviations.

- A figure should be made regarding current therapies, marketed and in development, against Carbapenem-resistant Acinetobacter baumannii Isolates, and their limitations;

Response: This important information has been summarized very nicely in a recently published review article by Shields et al. A sentence has been added to the introduction referring to this article (Reference #11).   

- The fact that the studies were only done in vitro, and not in vivo, or in a clinical setting, should be mentioned and discussed as limitations of the study;

Response: It is true that we are not presenting in vivo data but we have indicated in the title that the manuscript is presenting in vitro data, therefore not presenting in vivo data is not a limitation. In vivo data have been generated and published by other researchers.

- In spite of the promising results, the authors still found bacterial resistance in specific cases; strategies to overcome this should be discussed, such as, for example, drug molecule functionalization/modification;

Response: Speculating about potential drug molecule modification is beyond the scope of this paper and also beyond the expertise of a clinical microbiologist.

- The methods section should have a subsection describing the used statistical software and respective methodology;

Response: We have not applied statistics nor used statistical software or presented statistical data in our manuscript

- Given the general characteristics of the most promising molecules (lipophilicity/hydrophilicity, vulnerability do metabolism, etc.), what types of formulations would be best - Conventional? Nanosystems? And why; also what administration routes would be best?

Response: Cefiderocol is a licensed product that is available for intravenous application only. Speculating about potential alternative formulations and administration routes is beyond the scope of this paper.